# Critical Roles of the Cysteine–Glutathione Axis in the Production of γ-Glutamyl Peptides in the Nervous System

**DOI:** 10.3390/ijms24098044

**Published:** 2023-04-28

**Authors:** Junichi Fujii, Tsukasa Osaki, Yuya Soma, Yumi Matsuda

**Affiliations:** 1Department of Biochemistry and Molecular Biology, Graduate School of Medical Science, Yamagata University, Yamagata 990-9585, Japan; 2Graduate School of Nursing, Yamagata University Faculty of Medicine, Yamagata 990-9585, Japan

**Keywords:** xCT, glutathione peroxidase, glutaredoxin, γ-glutamyl transferase, amino acid transporter, calcium-sensing receptor

## Abstract

γ-Glutamyl moiety that is attached to the cysteine (Cys) residue in glutathione (GSH) protects it from peptidase-mediated degradation. The sulfhydryl group of the Cys residue represents most of the functions of GSH, which include electron donation to peroxidases, protection of reactive sulfhydryl in proteins via glutaredoxin, and glutathione conjugation of xenobiotics, whereas Cys-derived sulfur is also a pivotal component of some redox-responsive molecules. The amount of Cys that is available tends to restrict the capacity of GSH synthesis. In in vitro systems, cystine is the major form in the extracellular milieu, and a specific cystine transporter, xCT, is essential for survival in most lines of cells and in many primary cultivated cells as well. A reduction in the supply of Cys causes GPX4 to be inhibited due to insufficient GSH synthesis, which leads to iron-dependent necrotic cell death, ferroptosis. Cells generally cannot take up GSH without the removal of γ-glutamyl moiety by γ-glutamyl transferase (GGT) on the cell surface. Meanwhile, the Cys–GSH axis is essentially common to certain types of cells; primarily, neuronal cells that contain a unique metabolic system for intercellular communication concerning γ-glutamyl peptides. After a general description of metabolic processes concerning the Cys–GSH axis, we provide an overview and discuss the significance of GSH-related compounds in the nervous system.

## 1. Introduction

Methionine (Met) and cysteine (Cys) are the only sulfur-containing amino acids, and Cys can be synthesized through the transsulfuration reaction associated with Met metabolism. Met is metabolized to S-adenosylmethionine (SAM), which provides either the carbon backbone for polyamines or methyl groups for some other compounds, including DNA [1]. Meanwhile, Cys comes from extracellular sources or is the product of the transsulfuration reaction and then becomes the precursor for sulfur-containing components or mediators. These include glutathione (GSH), taurine, coenzyme A, hydrogen sulfide, iron–sulfur [Fe–S] cluster, and persulfides [2,3]. Although Cys is cytotoxic at high concentrations, because of the extreme demand, the cellular levels of Cys remain low, approximately 100-fold lower than that of GSH in astrocytes [4].

GSH maintains the redox state in cells by acting as a major antioxidant, and detoxifies xenobiotics through GSH conjugation. GSH also acts as a donor substrate of a γ-glutamyl group and a carrier for delivering Cys. The reactivity of GSH with reactive oxygen species (ROS) is marginal and, hence, the antioxidant action is largely attributed to its ability to donate an electron to glutathione peroxidases (GPX) [5]. GSH is also involved in protecting proteins from oxidative insult via a glutaredoxin (GRX)-involved reaction [6]. Moreover, glutathione S-transferase (GST) catalyzes GSH conjugation reactions of xenobiotics, resulting in the excretion of products as GSH conjugates [7].

In fact, the GSH system and the thioredoxin (TRX) system are two major redox systems that protect the animal body, including the central nervous system, from oxidative damage [8,9]. GSH plays a vital role in neural physiology but also exerts a unique function in neuroprotection [10,11]. Hence, the GSH system is a promising drug target for several neurological disorders [12,13,14]. In this review article, we first reviewed the metabolic pathways and molecular dynamics of GSH and then provided an overview of significant actions of GSH in maintaining redox balance and modulating extracellular signaling in the neuronal system. We also reviewed the origin and function of γ-glutamyl peptides that occur in close association with GSH metabolism.

## 2. Regulation of Cellular GSH Content by Balancing Synthesis, Recycling, and Degradation

GSH is present in the mM range in most mammalian cells. Because GSH is rapidly consumed through oxidation, conjugation, or degradation, cellular levels are mainly maintained by the synthesis of GSH from constituent amino acids glutamate (Glu), Cys, and glycine (Gly) (Figure 1). Indeed, Cys availability generally determines the GSH content in cells [4]. Here, we provide an outline of GSH metabolism, followed by highlighting recent advances in our knowledge of GSH homeostasis that are necessary for understanding its role in the nervous system. For a further understanding of the systemic roles of genes responsible for the GSH system or recycling in vivo, readers are referred to reviews on mouse model of GSH deficiencies caused by the ablation of genes for γ-GCS, GSS, and GGT, as well as GSR [15,16].

### 2.1. Dynamics of Cys Responsible for Cellular Redox Homeostasis

It is generally understood that GSH is not taken up by cells in the intact form, and recent studies indicate that GSH transporters are present in certain cells, such as endothelial cells [17,18]. GSH is enclosed in exosomes, which may act as intercellular vehicles across the blood–brain barrier [19]. Nevertheless, it remains true that extracellular GSH does not fulfil cellular requirements, and that most cells need to actively synthesize GSH from constituent amino acids in order to maintain the required levels. Cys and GSH share many chemical properties, notably in redox reactivity to maintain cellular homeostasis. Cys is present only in the submillimolar range due to its cytotoxicity and rapid consumption, but GSH is a safe form and, hence, can be present in several millimolar concentrations. This makes GSH a dominant player in cellular redox reactions compared to Cys. In the meantime, Cys plays unique roles, as represented by its serving as a direct precursor for sulfur-containing molecules that include taurine, cysteamine, coenzyme A, and [Fe–S] cluster. For example, taurine is the most abundant amino acid that is present in the free form, which is ~50 mM in hepatocytes and the brain [4,20], and the levels of coenzyme A are approximately 2.3 mM in mitochondria [21]. Therefore, the cellular levels of Cys dynamically change upon stimuli and often become insufficient. To meet this need, Cys is either actively taken up from the extracellular milieu or synthesized via the transsulfuration pathway.

Met metabolism is initiated by reacting with ATP, which results in the formation of SAM with the releases of a phosphate and pyrophosphate [2]. SAM functions as a methyl group donor for the synthesis of creatinine and adrenalin and also for the methylation of DNA and histones, which act as the epigenetic regulation of gene expression. The resulting S-adenosylhomocysteine (SAH) becomes homocysteine (Hcy) by releasing adenosine. In the transsulfuration pathway, cystathionine β-synthase (CBS) combines Hcy with serine (Ser) to form cystathionine (Cst) [22]. Cystathionine γ-lyase (CSE) then cleaves Cst to Cys and 2-oxobutyrate. CBS and CSE are also known to catalyze the conversion of Cys to hydrogen sulfide [23]. The activity of the transsulfuration pathway varies depending on the types of cells. For example, the pathway produces approximately 50% of the Cys that is needed for the synthesis of GSH in the liver [24], so that primary cultured hepatocytes can survive for more than one week under cultured conditions in the absence of cystine [25]. The transsulfuration pathway also actively provides Cys for GSH synthesis in the brain, notably in astrocytes [26]. Nevertheless, depletion of Cys causes ferroptosis in many cells in culture systems [27], which implies that the intrinsically produced Cys from the transsulfuration pathway does not fulfil the need in such cells. Limited contribution of the transsulfuration pathway in some organs could increase reliance on the blood-mediated supply of Cys for GSH synthesis.

### 2.2. Uptake of Extracellular Cys and Cystine

Cells take up Cys via neutral amino acid transporters (NAATs), examples of which are the alanine serine cysteine transporter (ASCT) and the L-type amino acid transporter 2 (LAT2), whereas neurons employ a neutral amino acid transporter excitatory amino acid carrier type 1 (EAAC1, also called EAAT3) (Figure 2) [28,29]. For a more detailed review of this, readers are referred to a report on the relationship between GSH synthesis and EAAC1 in neurons [30]. Cystine, a Cys dimer linked by a disulfide bridge, is the predominant form in extracellular fluid, notably under cultivation conditions. System b^0,+^ acts as a cystine transporter, but its main component, b^0,+^ AT, is expressed exclusively in the kidney. Another cystine transporter, xCT (the protein encoded by *SLC7A11*), which shows no structural similarity to b^0,+^ AT, is constitutively expressed in limited organs, such as the brain and the immune system [31,32]. In situ hybridization indicates the characteristic expression of xCT in the brain, which suggests exchanging cystine and Glu in the central nervous system is physiologically significant [33]. The presence of xCT has been demonstrated in blood/brain/CSF interface areas and in an astrocyte subpopulation [34]. xCT transports not only cystine but also cystathionine [35], which can be converted to Cys via the transsulfuration pathway in competent cells. If cells do not express *SLC7A11* and hence are incapable of taking up cystine, *SLC7A11* expression can be induced in response to oxidative conditions, likely through activation of Nrf2 and ATF4 [36,37]. In fact, in most cells that are cultivated under conventional conditions, the expression of *SLC7A11* is required for the uptake of extracellular cystine to occur. Cystine inside cells is then reduced to Cys by TRX1 and the TRX-related protein 14 kDa (TRP14) [38].

The *SLC7A11*-knockout (KO) mouse shows a redox imbalance in blood plasma, but otherwise a normal phenotype, probably because *SLC7A11* is expressed only in limited organs [39]. Once isolated from an *SLC7A11*-KO mouse, however, fibroblasts die if there is no supplementation of reductants. This type of cell death is referred to as ferroptosis, which is also typically induced by the inhibition of xCT or GPX4 [40]. *SLC7A11* expressed in glia has been reported to regulate hippocampal synaptic strength [41]. Whereas astrocytes take up cystine via xCT, neurons take up Cys only via EAAC1 [42,43]. Due to the pivotal function of both Cys and Glu in the brain, many studies have involved the role of xCT concerning neurodegenerative diseases, such as schizophrenia, inflammation, and tumor [44,45]. Whereas cystine uptake is a major concern for non-neuronal cells, Glu, a counter amino acid of cystine, is a major excitatory neurotransmitter. The concern is that excessive activation of the N-methyl-D-aspartate (NMDA) receptor by extracellular Glu can cause oxidative Glu toxicity [46,47]. Whereas both activated and resting microglial cells express *SLC7A11* and release Glu via xCT, neither changes in GSH content nor neuronal dysfunctions are observed in xCT-KO mice [48]. However, Glu exported via xCT triggers the removal of postsynaptic AMPA receptors and suppresses glutamatergic synapse strength [41]. Whereas *SLC7A11* is not expressed in motor neurons, *SLC7A11* expression is induced in activated microglia in amyotrophic lateral sclerosis (ALS) mice carrying mutant *SOD1*, the gene encoding cytosolic superoxide dismutase (SOD) [49]. The deletion of *SLC7A11* leads to the premature onset of ALS symptoms, but the disease phase then progresses slowly, leading to the survival of more motor neurons in this model mouse.

Regarding tumors, the stable expression of *SLC7A11* appears to promote the growth of gastrointestinal cancer cells and enhance malignancy by increasing the redox potential in in vivo conditions [50]. Gliomas are the most common malignant tumor in the central nervous system. The inhibition of xCT reportedly abrogates glioma-induced neurodegeneration and brain edema [51]. The EGF receptor is associated with higher levels of GSH due to the presence of xCT, and the inhibition of xCT suppresses the EGFR-dependent enhancement of antioxidant capacity in glioma cells [52]. Results reported in many studies indicate that the overexpression of *SLC7A11* is associated with a glioma malignancy, thus making xCT one of the potential targets for the treatment of gliomas [53,54]. Apoptosis-resistant tumors tend to be sensitive to ferroptotic stimuli [55,56]. Accordingly, xCT has attracted considerable attention in cancer biology, since the induction of ferroptosis by targeting xCT appears to be a promising approach for the treatment of apoptosis-resistant tumors [57,58,59]. Contrary to these observations, however, the overexpression of *SLC7A11* induces cell death in glioblastomas under conditions of glucose deprivation caused by high-density cultivation [60]. Surviving glioblastoma cells at high density undergo inactivation of mammalian targets of rapamycin-1 (mTOR-1) and xCT degradation by lysosomes in order to avoid ferroptosis. The activated form of the antitumor protein p53 induces apoptosis [61], but also ferroptosis [62]. In this case, xCT is downregulated by p53, which leads to ferroptosis due to GSH depletion [58]. Other than genetic regulation, the activation of p53 promotes ferroptosis by means of differential mechanisms [62,63]. However, a complication is that p53 may induce the expression of *SLC7A11* under certain circumstances, leading to the suppression of ferroptosis [64]. It appears that clarification of such an inconsistency is needed.

### 2.3. GSH Synthesis and Consequence of Its Defect

Cellular GSH levels are maintained mainly through de novo synthesis from constituent amino acids via the coordinated action of γ-glutamyl–cysteine synthetase (γ-GCS) and glutathione synthetase (GSS). γ-GCS is a heterodimeric enzyme composed of a catalytic subunit that is encoded by the *GCLC* gene and a modifier subunit that is encoded by the *GCLM* gene. The inhibition of GCLC by a specific inhibitor buthionine sulfoximine decreases GSH to negligible levels within one day [65]. γ-GCS ligates the carboxyl group of the side chain of Glu to the amino group of Cys, resulting in the formation of γ-glutamyl–cysteine (γ-Glu-Cys). GSS then adds a glycine unit to γ-Glu-Cys, which completes the synthesis of tripeptide γ-Glu-Cys-Gly, which is referred to as GSH. Because Cys is the preferred acceptor amino acid of the γ-GCS reaction (K_m_ = 0.1–0.3 mM), ordinary amounts of Cys enable the coupled reaction of γ-GCS and GSS, which results in the production of GSH. However, under a Cys deficiency, γ-GCS catalyzes the γ-glutamylation of other amino acids due to its low specificity toward the acceptor substrate, leading to the production of a variety of γ-glutamyl peptides depending on amino acid status, even in in vivo conditions [65].

Defects in genes for GSH-synthesizing enzymes are few in humans, probably due to the pivotal roles of GSH [66]. γ-GCS and GSS deficiencies are commonly associated with a decrease in GSH levels, hemolytic anemia, and sometimes neurological dysfunction, although a deficiency of GSS is more frequent than that of γ-GCS [67,68,69]. GSS-deficient patients show 5-oxoprolineuria, hemolytic anemia, and neurological dysfunction [70,71]. Fibroblasts from a GSS-deficient patient have been reported to show increased levels of cysteine and γ-Glu-Cys but decreased levels of GSH [72].

Mice with the knockout of these genes have been employed to examine the function of GSH in vivo. *GLCL*-null mutant mice die by embryonic day 7.5 due to accelerated cell death [73,74]. Supplementing culture media with GSH or N-acetylcysteine (NAC), a precursor for Cys and complement to the synthesis of GSH, rescues fetal embryonic fibroblasts (MEF) isolated from *GCLC*-KO mouse embryos from this fatal nature, which demonstrates that a decline in GSH is the cause of death. The hepatocyte-specific ablation of *GCLC* results in the development of steatosis, mitochondrial injury, and elevated lipid peroxidation [75]. Mice with Treg-specific ablation show increased Ser metabolism, mTOR activation, and proliferation [76]. Thus, GSH appears to restrict the availability of Ser, which results in the Treg function being preserved. Myeloid-specific deletion of *GCLC* compromises the activation of mTOR-1 and the expression of the *c-Myc* transcription factor, which abrogates energy utilization [77]. The conditional knockout of *GCLC* in neuronal tissues induces brain atrophy which is accompanied by neuronal loss and neuroinflammation and appears to be associated with the activation of microglia [78]. On the other hand, no apparent phenotypes have been observed in *GCLM*-KO mice, although the levels of GSH in some organs are 9–16% that of wild-type mice [79]. MEF isolated from *GCLM*-KO mice are more sensitive to oxidants, such as hydrogen peroxide and arsenic [80]. An acetaminophen overdose aggravates liver damage in *GCLM*-KO mice but can be rescued by the administration of NAC [81]. On the contrary, a transgenic mouse overexpressing *GCLM* shows an increased resistance to acetaminophen-induced liver damage [82] and the hydrogen peroxide-induced breakage of single-strand DNA, which are consistent with the pivotal roles of GSH in antioxidation. *GCLM*-KO fibroblasts show an increase in intracellular ROS levels, and this results in diminished cellular proliferation and increased senescence [83]. Concerning the central nervous system, social isolation causes neurochemical alterations, such as an elevation in the concentrations of N-acetylaspartate, alanine, and glutamine, which may be attributed to decreased GSH levels [84]. Several metabolic abnormalities are observed in the livers of alcohol-treated *GCLM*-KO mice [85], although its physiological significance has not been examined any further.

### 2.4. Consumption of GSH and Reductive Recycling of GSSG

Cellular GSH is depleted mainly thorough the secretion of forms of oxidized glutathione (GSSG), glutathione S-conjugates, and S-nitrosoglutathione (GSNO). The ATP-binding cassette (ABC) transporter protein superfamily that contains 49 members in humans [86] is responsible for the secretion of various compounds with endogenous or exogenous sources [87]. The multidrug resistance regulator (MRP) subfamily, which is classified into the ABC transporter subfamily C, mediates the export of GSH-related compounds [88]. Among the MRP members, the transport of GSH and glutathione conjugates by MRP1 (gene symbol; ABCC1) has been extensively characterized [89]. Other forms of MRPs, MRP2 to MPR8 and CFTR (ABCC7, cystic fibrosis transmembrane conductance regulator), also transport some glutathione-related compounds.

The reductive recycling of GSSG is catalyzed by the action of glutathione reductase (GSR), which uses NADPH as an electron donor. GSR plays a crucial role in cell survival and is induced in an Nrf2-dependent manner under conditions of oxidative stress [90]. GSR is inhibited by 1,3-bis(2-chloroethyl)-1-nitrosourea (BCNU), an anticancer agent. It should also be noted that testes express a novel enzyme, thioredoxin/glutathione reductase, that reductively recycles both oxidized TRX and oxidized GSH as substrates [91] and is mainly associated with sperm maturation [92,93].

A deficiency of GSR is very rare in humans, and only 18 mutations in the GSR gene have been reported by 2022 [94]. Mutations in GSR appear to cause changes in the structure and function of the GSR protein, which may be associated with obstructive heart defects and hereditary anemia. Several mice with defects in the GSR gene have now been established. The *Gr1^a1Neu^* mouse, which is generated by treating it with a mutagen, isopropyl methanesulfonate, exhibits less than 10% GSR activity [95]. A frameshift in exon 6 results in a premature stop codon in exon 7 and leads to the production of a dysfunctional GSR protein [96]. Treatment with diquat, a redox cycling toxicant, causes a renal proximal tubule injury [97], but an acute lung injury from continuous exposure to 95% oxygen is rather alleviated in these mice [98]. *GSR*-KO mice have also been generated by the gene-targeting technique and found to be less sensitive to the effects of hyperoxia [99]. Since the upregulation of TRX is observed, cytoprotective responses supported by the TRX system appear to compensate for the GSR deficiency. Mice with a double knockout of *GSR* and thioredoxin reductase-1 remain viable, but the continuous synthesis of GSH is required, probably due to the inability to reduce GSSG [100]. Since Met fuel supports the survival of the double knockout mice, the transsulfuration pathway coupled with Met metabolism may supply Cys for the continuous synthesis of GSH.

### 2.5. GGT Responsible for γ-Glutamyl Peptide Metabolism on the Cell Surface

Hgt1p is a high-affinity GSH transporter in *S. cerevisiae*, but no orthologous gene is known in mammals [17,101]. Accordingly, it is generally thought that GSH does not enter cells in its native form. The γ-glutamyl moiety of GSH provides protection against proteolytic degradation, and, hence, the removal of the γ-glutamyl moiety from GSH by GGT is essential for the incorporation of the remaining Cys–Gly dipeptide or amino acids after degradation by a dipeptidase in the extracellular milieu [102]. GGT on the cell surface catalyzes either the hydrolysis or transfer of the γ-glutamyl moiety from GSH to amino acids or peptides. Since the human genome contains several *GGT*-related genes along with pseudogenes, a systematic nomenclature has been applied to the *GGT* gene family [103]. Circulating GSH in the bloodstream is rapidly degraded by GGT1 on the luminal surface of the renal brush border membrane [104]. The increased expression of *GGT* is a well-established biomarker for some malignant tumors. Whereas GGT is originally anchored to the plasma membrane of cells, proteolytic cleavage increases the release of the active enzyme, which enables its detection in blood by measuring the activity. Due to its clinical significance, numerous studies have been performed on members of the GGT family from the standpoints of genetics, protein chemistry, and enzymology [105].

GGT either removes a γ-glutamyl moiety by hydrolysis or transfers it to amino acids or peptides, thus resulting in the formation of a variety of γ-glutamyl peptides [104]. The brush border membrane in the kidney dominantly expresses *GGT1*, which is the best-characterized form. After glomerular infiltration, GSH undergoes degradation by GGT1, and the resulting Cys–Gly dipeptide or amino acids are largely taken up by the corresponding transporters, which include the H^+^-coupled peptide transporter PEPT2 [106,107] and neutral amino acid transporters (NAATs), such as ASCT1 and ASCT2 [31]. The physiological benefit of transferring the γ-glutamyl moiety has long been a subject of debate. Given the fact that Glu is toxic in the neuronal system, it has been speculated that transferring a γ-glutamyl moiety rather than its release as Glu would decrease its toxic effect on neurons [108]. Moreover, whereas GSH is incorporated at very low levels, there appear to be transporters for γ-glutamyl peptides or receptors as described below.

In the brain, *GGT7* is expressed by pericytes and perivascular astrocytes [109]. *GGT7* is frequently downregulated in gastric cancer due to the methylation of the promoter sequences of the gene [110]. GGT7 directly binds the mitophagy regulator RAB7, and it is then translocated from the nucleus to the cytoplasm, which leads to mitophagy by increasing its mediators/inducers. Consequently, the mitotic signaling of the mitogen-activated protein kinase (MAPK) is suppressed. Thus, GGT7 appears to act as a tumor suppressor. Consistent with this notion, glioblastoma patients with low levels of *GGT7* expression show elevated levels of ROS and exhibit poor prognoses [111]. The expression of *GGT1* also appears to be involved in the determination of the sensitivity of glioblastoma cells to cystine deprivation-induced ferroptosis in cases of high cell density [112]. Since the results were obtained from the cell cultivation alone, an in vivo study would be required if this regulation of ferroptosis by GGT1 is to be considered a target of cancer treatment.

Patients with a GGT deficiency show low GGT activity and develop glutathionuria with increased plasma GSH levels as well as the presence of γ-Glu-Cys and Cys in the urine [113]. A mutant mouse that has spontaneous *dwg* and *dwg (Bayer)* mutations in *GGT1* shows phenotypic abnormalities [114]. *GGT^enu1^* mice with a point mutation within the protein-coding region of *GGT1* have been generated by treatment with N-ethyl-N-nitrosourea [115]. The phenotypes of the mutant mice are associated with glutathionemia, glutathionuria, and growth retardation, which are partially rescued by a cysteine prodrug, L-2-oxothiazolidine-4-carboxylate (OTZ) [116]. Genetically modified mice have been established by the targeted disruption of *GGT1* [117] or *GGT5* [118]; however, no specific abnormalities in the central neuronal system have been reported for these mice so far.

### 2.6. γ-Glutamylcyclotransferase Responsible for Intracellular GSH Degradation

The γ-glutamyl group is protective against the peptidase-mediated degradation of GSH, but γ-glutamylcyclotransferase, which is now recognized as an intrinsic member of the ChaC family of proteins, initiates its degradation by the removal of the γ-glutamyl group [102]. ChaC1 and ChaC2 are involved in the cytosolic degradation of GSH with a relatively high K_m_ (~2 mM) by the hydrolytic removal of the γ-glutamyl moiety with the production of 5-oxoproline and a Cys–Gly dipeptide [119]. Whereas the enzymatic reaction is the same, ChaC1 shows an approximately 20-fold higher activity than that for ChaC2 [102,120]. There are also differences in gene regulation; *ChaC1* is upregulated by stress, whereas *ChaC2* is constitutively expressed.

The expression of ChaC1 is sustained at minimal levels but is induced upon amino acid starvation and endoplasmic reticulum (ER) stress via an ATF4-CHOP cascade [121]. ChaC1 is also regarded as a potential marker for ferroptosis [122]. The induced expression of *ChaC1* is associated with the accelerated degradation of intracellular GSH, which may result in the development of ferroptosis by suppressing the GPX4-meidated detoxification of lipid peroxides [123]. Approximately 50 percent of early-age-at-onset cases of Parkinson’s disease (PD) have been linked to bi-allelic mutations in genes encoding DJ-1, Parkin, and PINK1. DJ-1 protects against the development of PD because a mutation in the gene is the cause of autosomal recessive early-onset PD [124]. The knockout of *DJ-1* results in the upregulation in the expression of *ChaC1*, which indicates that DJ-1 is involved in maintaining GSH by suppressing the expression of ChaC1 through inhibiting the activation of the transcription factor ATF3 [125].

ChaC2 is highly enriched in undifferentiated human embryonic stem cells, and its downregulation decreases the levels of GSH and blocks their self-renewal [126]. The knockdown of *ChaC1* restores the self-renewability of *ChaC2*-downregulated cells, suggesting that ChaC2 competes with ChaC1 in the maintenance of GSH homeostasis. Thus, ChaC1 and ChaC2 show the same enzymatic activity toward GSH, but they appear to compete in GSH metabolism, i.e., ChaC1 depletes cellular GSH, but ChaC2 functions to maintain it. In terms of tumors, the expression of *ChaC2* is frequently downregulated in gastric and colorectal cancers, suggesting that it is a tumor suppressor [127]. Meanwhile, breast cancer [128] and hepatocellular carcinoma [129] patients with elevated *ChaC2* show poor prognoses. It is a rather general phenomenon that the antioxidant system suppresses tumorigenesis, but the acquisition of the system tends to make tumors resistant to chemotherapy and become more malignant.

### 2.7. Dipeptidase for Recruit of Cys from GSH Degradation Product

The removal of the γ-glutamyl moiety from GSH by GGT on the cell surface produces a Cys–Gly dipeptide, which can be taken up by the peptide transporter PEPT2 expressed in glia, but not in neurons of the brain [130] nor in the peripheral nervous system [131]. We recently identified carnosine dipeptidase (CNDP) 2 as a protein that is elevated in primary macrophages from an *SLC7A11*-KO mouse [132]. CNDP1 and 2 belong to a family of dipeptidases for carnosine (β-alanyl-histidine) [133]. Whereas CNDP1 specifically degrades carnosine, CNDP2 is more specific for the Cys–Gly dipeptide. The recycling of Cys from Cys–Gly by CNDP2 would support GSH synthesis by supplying Cys, thereby suppressing oxidative stress by promoting the activity of GPX. The knockout of *CNDP2*, however, did not cause much change in the Cys–Gly dipeptidase activity, which suggests the dominant presence of other enzymes with similar substrate specificities or compensation by other enzymes that may not be specific to Cys–Gly dipeptide [132].

CNDP2 is upregulated in several tumors and appears to act as a tumor suppressor in hepatocellular carcinomas [134] and in gastric cancer [135], which appears to be achieved by stimulating MAPK, notably p38 and JNK, thereby inducing apoptosis. Meanwhile, CNDP2 may stimulate the growth of colon cancer [136] and ovarian cancer cells [137], suggesting tumorigenic action. When tumor cells gain the ability to overexpress *CNDP2*, the elevation in GSH content would render tumor cells resistant against anticancer agents. In this regard, the inhibition of CNDP2 activity appears to be beneficial for anticancer therapy. For example, an inhibitor for CNDP2, ubenimex, also known as bestatin, is an anticancer agent for adult acute nonlymphocytic leukemia. Hence, the anticancer effects of ubenimex may be partly attributed to the inhibition of CNDP2 [138,139].

## 3. GSH Protects Cells against Stress through Multiple Pathways

GSH is ubiquitously present in cellular organelles and exhibits pleiotropic functions that are largely associated with the sulfhydryl (SH) group in the Cys residue [140]. Superoxide radicals, which are produced by one-electron donation to molecular oxygen under oxidative stress, are converted to hydrogen peroxide by SOD. Then, electron donation from GSH to GPX effectively results in the reduction of cytotoxic peroxides [5]. GSH protects proteins from oxidative damage and allows the native structure of proteins to be maintained through the action of GRX [141]. The conjugation of xenobiotics with GSH, which is catalyzed by the action of GST, accelerates their excretion [142].

### 3.1. Reductive Detoxification of Peroxides via GPX

The oxidation of GSH occurs by the direct reaction with oxidants, but this is not a very efficient process. The reduction of peroxides to the corresponding alcohols is effectively catalyzed by GPX, which results in GSSG [5]. Whereas certain other proteins, such as catalase and peroxiredoxin, exhibit peroxidase activities, GPX is a dominant peroxidase family that is made up of eight members in mammals. Because GPX is constitutively active within cells and requires GSH other than substrate for activity, the supply of GSH determines the GPX activity. GPX1 to GPX4 possess a selenocysteine (Sec) instead of a Cys in their catalytic center (Figure 3). Replacing Sec with Cys decreases the catalytic efficiency of this molecule by less than 1%. Whereas Sec-containing wild-type GPX4 is capable of conferring resistance to irreversible overoxidation by peroxides [143], excessive hydrogen peroxide inactivates GPX1 by oxidatively converting Sec to dehydroalanine in human erythrocytes [144]. A diet with selenium deficiency results in a decrease in their activity due to the impaired synthesis of Sec. Whereas GPX largely functions to reduce hydrogen peroxides by means of the donation of an electron from GSH, GPX4 reduces phospholipid hydroperoxides to their alcohol forms and, hence, can suppress ferroptosis [27,40]. Since ferroptosis is considered to occur under a variety of physiological and pathological conditions, the suppression of lipid peroxidation by GPX4 has attracted much attention [145,146]. The genetic ablation of GPX4 causes embryonic lethality in mice, which can be overcome by vitamin E (α-tocopherol) supplementation [147], whereas this is not the case in mice lacking other *GPX* genes. Peroxiredoxin (PRDX) is another large family of proteins with peroxidase activity. PRDX family members mostly exhibit thioredoxin-dependent peroxidase activity, whereas PRDX6 exceptionally exhibits GSH-dependent peroxidase activity towards lipid peroxides. Thus, the enzymatic properties of PRDX6 are similar to those of GPX4, but there is essentially no similarity in their structure [148]. Since numerous studies have reported on the roles of GPX in neuronal systems, readers are referred to other review articles for details concerning the reactions and physiological significance of the GPX family [5,146,149]. The decline in the GPX activity increases peroxides, which, in the presence of iron, results in the production of hydroxyl radicals that cause oxidative DNA damage. Then, cells with DNA damage may undergo tumor development [27]. Oxidative stress-induced tumorigenesis is a major research subject in itself and is discussed repeatedly, so we will not discuss it further in this paper.

### 3.2. Rescuing Proteins from Oxidative Modification through GRX

Whereas SH groups can exist in four oxidized states: disulfide, sulfenic acid, sulfinic acid, and sulfonic acid, under an oxidative environment, the disulfide and sulfenic acid forms can generally be reduced by a biological redox system [16]. Cys–SH groups of proteins can form mixed disulfides with their own SH group or with SH groups from other molecules under oxidizing conditions (Figure 4A). Since GSH is the most abundant thiol, a mixed disulfide with GSH, called S-glutathionylation, occurs preferentially. This modification either inhibits or activates the enzyme, depending on the position of the Cys residue within the protein [6]. Approximately 1% of total glutathione is present as the mixed disulfide form with proteins in the normal liver, and the amount of protein-bound glutathione reaches 20–50% upon an oxidative insult [150]. The glutathione thiyl radial and GSNO form the S-glutathionylation of proteins in an accelerated manner. S-glutathionylation proceeds preferentially under oxidative conditions, but is not always harmful, because the formation of a mixed disulfide prevents further oxidation of the SH group [151]. GRX has been identified as a GSH-dependent reductase of disulfides in ribonucleotide reductase during its catalytic cycle [152,153] and, hence, acts as an alternate electron donor for TRX [6]. Because S-glutathionylation protects Cys–SH groups in proteins from further oxidation and since they can be reduced back, this post-translational modification is regarded as a type of protective mechanism for essential SH groups in proteins under oxidative stress [141]. GRX can reduce proteins with intrinsic disulfide bonds or S-glutathionylated proteins back to the native conformation by employing GSH, which releases GSSG. Mammals produce two GRXs, GRX1 and GRX2 [141,153]. Whereas GRX1 resides mainly in the cytoplasm, it can be translocated into the nucleus upon certain stimuli. GRX2 is present in two forms, namely, a mitochondrial and a nuclear GRX.

Redox regulation by the GSH/GRX system plays a pivotal function in the central nervous system [154,155]. The excitatory amino acid L-β-*N*-oxalylamino-L-alanine (L-BOAA) causes corticospinal neurodegeneration in humans and the loss of GSH in mice. The mitochondrial electron transfer complex (ETC)-I in the motor cortex is selectively lost by L-BOAA, but GRX appears to protect ETC-I from degradation [156]. Estrogen may be involved in preserving higher levels of GRX in certain regions of the central nervous system and, in female mice, protects them against mitochondrial dysfunction caused by L-BOAA [157]. GRX1 helps to maintain mitochondrial integrity and prevents the loss in mitochondrial membrane potential caused by L-BOAA [158]. 6-Hydroxydopamine is easily oxidized and results in the formation of the quinone form, which is a highly reactive species and a powerful neurotoxin [159]. Both the TRX and the GRX systems directly mediate the reductive detoxification of 6-hydroxydopamine quinone and protect neurons from dopamine-induced cell death.

GRX1 reportedly regulates the protein levels of DJ-1 in the midbrain of mice [160]. The mitochondrial form of GRX2 as well as TRX1 contributes to neuronal integrity during hypoxia [161] and protects neuronal cells against 1-methyl-4-phenyl-1,2,3,6-tetrahydropyridine (MPTP)-mediated mitochondrial dysfunction [162]. The downregulation of GRX1 leads to dopaminergic degeneration and PD-relevant motor deficits in mice [163]. Exogenously added cell-permeable PEP-1-GLRX1 also suppresses the dopaminergic neuronal cell death induced by MPTP [164]. GRX2 may also be involved in the function of mammalian dopaminergic cells and oligodendrocytes through the biogenesis of [Fe–S] clusters [165,166]. Increased levels of GRX1 are associated with the early onset of PD in patients, which suggests that upregulated GRX1 promotes neuroinflammation and leads to the development of PD [167]. However, other studies have reported decreased levels of GRX in PD patients [168]. Given the roles of GRX in protein thiol homeostasis, the upregulation of GRX1 is rather considered to be result of a compensatory reaction against oxidative protein modification.

The oxidation of F-actin caused by amyloid-β (Aβ) is diminished by GRX1 in Alzheimer’s disease (AD) model mice [169]. A decrease in GRX1 levels may lead to synaptic dysfunction during AD pathogenesis by directly disrupting the F-actin architecture in spines. On the contrary, Aβ may also exert neurotoxicity in AD through oxidizing GRX1 or TRX1 [170]. In familial ALS patients, the aggregation of mutant SOD1 is a proposed cause for the degeneration of motoneurons. Whereas the overexpression of GRX1 increases the solubility of mutant SOD1 in the cytosol, this does not alleviate mitochondrial damage in SH-SY5Y cells. However, the overexpression of GRX2 increases the solubility of mutant SOD1 in mitochondria and preserves mitochondrial function, which results in neuronal cells being protected from apoptosis [171]. Thus, GRX, with the help of GSH, can prevent the development of major neurodegenerative diseases such as PD, AD, and ALS.

### 3.3. Glutathione Conjugation in the Detoxification of Xenobiotics and in the Production of Bioreactive Compounds

Conjugation with GSH, glucuronate, and sulfate are catalyzed by GST, UDP-glucuronate transferase, and sulfotransferase, respectively, and constitute three major detoxification systems for xenobiotics and intrinsic compounds that are highly hydrophobic [172,173]. Oxygen molecules are usually introduced into hydrophobic compounds by the action of cytochrome P450 oxidases (CYP) before glutathione conjugation by GST (Figure 4B). In addition to GSH conjugation, GST also exhibits GSH-dependent peroxidase activity, albeit with much less efficiency compared to GPX. The mouse has 21 *GST* genes, and many of these genes are genetically knocked out, as reported in previous studies [174,175,176]. Many polymorphisms are found in *GST* genes that are likely risk factors for PD [177]. Whereas glucuronidation and sulfate conjugation are generally thought to have largely beneficial effects, excessive GSH conjugation may cause redox imbalance and damage cells due to the consumption of GSH, as is typically observed in the case of livers with an acetaminophen overdose [82].

On the other hand, GSH is used as a building block or cysteine donor for the synthesis of cysteinyl leukotrienes (CysLT) that include LTC_4_, LTD_4_, and LTE_4_ (Figure 5A). CysLT is the active component of a slow-reacting substance that causes anaphylaxis, the contraction of smooth muscle, and an increase in vascular permeability [178]. CysLT is also an inflammatory lipid mediator that is involved in the pathophysiology of respiratory diseases and may also be associated with defects in the central nervous system, including cerebral ischemia, epilepsy, and AD [179]. Regarding the synthetic pathway, 5-lipoxygenase first catalyzes the formation of an arachidonate epoxide, and GST family members (MGST2, 3, and GSTM4), as well as LTC_4_ synthetase, then catalyze the conjugation of GSH to the epoxide [180]. After their secretion from cells, GGT1 or GGT5 hydrolytically removes the γ-glutamyl group of LTC_4_, which results in the formation of LTD_4_. Extracellular peptidases finally catalyze the hydrolytic removal of the Gly unit from LTD_4,_ and this results in the production of LTE_4_. CysLT acts through G protein-coupled receptor subtypes that are referred to as CysLTR-1 and CysLT-2 and are present on neurons, astrocytes, microglia, and vascular endothelial cells in the brain [181]. Moreover, proteins, such as G protein-coupled receptor 17 (GPR17), G protein-coupled receptor 99 (GPR99), and peroxisome proliferator-activated receptor-γ (PPARγ), may also act as receptors for CysLT. Since CysLT is also involved in inflammatory responses, its excessive production may result in neuronal tissue damage.

### 3.4. GSNO in Nitric Oxide Signal Transduction

Nitric oxide (NO) that is produced through both enzymatic reactions and non-enzymatic reactions exerts a variety of beneficial functions, including the relaxation of the vasculature and the modulation of neurotransmission, whereas the presence of abundant levels may impair the redox balance [182]. Whereas the iron ion, notably ferrous iron, is the preferred target of NO, SH groups in amino acids and proteins are also reactive and are targets, which results in the formation of S-nitrosothiol (SNO). Since reactive SH tends to play a primary role in a redox reaction that includes TRX and the ubiquitin system [183], and non-protein thiols, including coenzyme A [184], S-nitrosylation is a pivotal post-translational modification that is involved in cellular signaling. Excessive nitrosylation of target proteins may cause dysfunction, aberrant activation of physiological processes, and ultimately cell death. S-Nitrosylation occurs in many different PD-related proteins, including peroxiredoxin 2, XIAP, and PDI [185]. The balance between S-nitrosylation and denitrosylation determines whether SNO acts as a signaling mechanism or causes nitrosative stress [186].

GSNO is dominantly produced due to the abundant presence of GSH in cells and may act as the donor for trans-nitrosylation reactions [187]. GSNO is transported out of cells via MRP [88]. GSNO-reducing activities, which are intrinsic to formaldehyde dehydrogenase (GSNOR), class III alcohol dehydrogenase (ADH5), and TRX/thioredoxin reductase, play pivotal roles in moderating GSNO levels [188]. ADH5 appears to be a major GSNO reductase that acts in an NADH-dependent manner [189]. Whereas carbonyl reductase 1 (CBR1) preferentially reduces GSNO [190,191], a form of aldehyde reductase AKR1A1 also exerts GSNO reductase activity with the formation of glutathione-sulfinamide derivatives [187]. A GSNOR deficiency induces the S-nitrosylation of focal adhesion kinase 1 (FAK1), which results in the enhanced autophosphorylation of FAK1 and tumorigenicity being sustained [192]. Since detailed information concerning the action of S-nitrosylation is not the focus of this review, readers are directed to recent review articles that are associated with neurodegenerative diseases [193,194].

### 3.5. GSH Status Associated with Neuronal Diseases

We briefly revisit the relationship between major neurological diseases and GSH status in this review. A GSH deficiency is associated with various neurological disorders, including neurodegenerative diseases, ischemic disease, schizophrenia, and tumors [59,195,196,197]. Ferroptosis, which is associated with declined GSH, is assumed to be involved in neurodegenerative diseases, including AD, PD, and ischemic disease [195,197,198]. Because polyunsaturated fatty acids (PUFA) are rich in the brain and are susceptible to being peroxidized, GPX4 is predominantly present and protects neurons from ferroptosis. Here we outline three typical types of neurodegenerative diseases, AD, PD, and ALS, in association with GSH metabolism.

AD is the most common neurodegenerative disease. Aggregation of the Aβ peptide eventually causes AD by inducing neuronal cell death. The hallmarks of AD include elevated ROS levels and the enhanced production of lipid peroxidation products, decreases in GSH and GPX4, and the accumulation of iron, which are also hallmarks of ferroptosis [199,200]. The administration of desferrioxamine, an iron chelator, to AD patients reportedly leads to a significant reduction in the rate of decline of daily living skills, suggesting that sustained iron chelation may be beneficial in slowing the progression of this disease [201]. The administration of α-tocopherol, which suppresses ferroptosis by inhibiting lipid peroxidation, was also reported to moderate AD in patients [202]. Moreover, treatment with NAC was reported to prevent cognitive impairment in an AD model mouse that was induced by intracerebroventricularly administered streptozotocin [203]. The injection of Aβ oligomers into the CA3 hippocampal region of the rat brain triggers synaptotoxic effects that are represented by abnormal Ca^2+^ signals and mitochondrial dysfunction, whereas feeding NAC for 3-weeks prior to Aβ injections prevented these deleterious effects [204]. It is therefore considered likely that ferroptosis is related to the onset or exacerbation of AD, and that the Cys–GSH axis exerts protective action in preventing neuronal cell death. Using *Drosophila* models, a close correlation between changes in GSH redox potential with AD disease onset caused by Aβ and progression was observed [205].

Dopamine is a pivotal neurotransmitter, and its deficiency is a cause for PD, which is also a common neurodegenerative disease, next to AD. Progressive dopaminergic neuronal loss in the substantia nigra pars compacta is the characteristic pathology of PD patients. Since iron accumulation is associated with PD, ferroptosis as well as oxidative stress are also considered to be possible causes [206,207,208]. Dopamine can be converted into dopamine-o-quinone in the oxidative pathway, which is efficiently catalyzed by the presence of metal ions such as iron, copper, and manganese, as well as by the action of ROS-producing enzymes, such as xanthine oxidase, cytochrome P450, prostaglandin H synthase, and lactoperoxidase [209]. The metabolism of GSH in association with dopamine is deeply involved in this fatal disease [210,211]. Several enzymes, including GST family members and LTC_4_ synthase, catalyze the nucleophilic addition of GSH to dopamine-o-quinone, which results in the formation of 5-S-glutathionyl dopamine (Figure 5B). MRP then appears to export 5-S-glutathionyl dopamine [86]. GGT that is present on the extracellular surface of astrocytes then catabolizes the removal of the γ-glutamyl group from 5-S-glutathionyl dopamine [212]. The Gly unit is finally removed by the action of an extracellular peptidase. The resulting Cys-dopamine is toxic to neurons in the substantia nigra pars compacta and may lead to neuronal death. In addition to the neuronal toxicity of these compounds, Cys/GSH consumption may impair GPX4 function and predispose neurons to ferroptosis. Similar processing occurs on glutathione-conjugated acetaminophen, and the resulting cysteinyl-acetaminophen is reportedly a major contributor to renal toxicity [213].

Despite the fact that it is much less widespread compared to AD and PD, ALS affects motor neurons in the cerebral cortex, brainstem and spinal cord and leads the death rate due to respiratory failure within five years. Oxidative cell damage also appears to be involved in the development of ALS, and, hence, the GSH redox system is considered to exert a beneficial action [214]. In fact, the modification of the Cys residue in either GSH or in proteins occurs and may be involved in the pathogenesis of ALS [215]. Whereas the significance of the Cys–GSH axis has been implied, clinical trials that include the administration of GSH, Cys or procysteine have not been successful. Meanwhile, the administration of edaravone, a radical scavenger, to rats was reported to alleviate spinal cord injury [216]. Edaravone is now a licensed radical-scavenging drug for the treatment of ALS as well as strokes [217,218]. These collective results suggest that radical scavenging rather than fueling the Cys–GSH axis might be more advantageous in slowing the progression of ALS.

### 3.6. Production of a Variety of γ-Glutamyl Peptides by Means of γ-GCS and GGT

A variety of peptides and amino acids, which may not always be composed of proteinous amino acids, are present in blood plasma and tissues including the brain (Figure 6). Taurine is synthesized from Cys with cysteine dioxygenase as the rate-limiting enzyme [219]. Astrocytes predominantly produce taurine, which exerts pleiotropic actions in the central nervous system [220]. The levels of taurine and 2-hydroxybutyrate, a metabolite of the transsulfuration pathway, are elevated in plasma and cells from sporadic ALS patients, suggesting that the pathogenesis of ALS is associated with metabolic stress [221]. Because taurine does not contain a carboxyl group, it presents largely in the free form and constitutes the second amino acid in dipeptides. γ-Glutamyl taurine (γ-Glu-Tau) is produced in the brain, and is reportedly mediated by GGT [222]. In fact, a variety of reactive peptides that include γ-glutamyl peptides have been reported to be present in the brain. We recently established a liquid chromatography–mass spectrometry (LC–MS)-based assay method that can provide structural information on the products of the enzymatic reaction of γ-GCS and GGT. Our results indicate that, despite the high K_m_ of taurine for the γ-GCS reaction, its abundant presence indeed also enables the generation of γ-Glu-Tau by γ-GCS [223,224]. The resulting γ-Glu-Tau may interact with excitatory aminoacidergic neurotransmission [225] and exert antiepileptic actions [226], although the nature of its function remains ambiguous [227]. GGT and γ-GCS appear to be the enzymes that are responsible for the production of various γ-glutamyl peptides, although the amount and diversity of γ-glutamyl peptides are small in the normal mouse brain compared to the liver and kidney [224]. These observations imply that the concentrations of Cys are properly maintained in the brain under healthy conditions. Aberrant syntheses of γ-glutamyl peptides other than GSH by γ-GCS and/or GGT reactions can also be a predictive marker for the condition of the central nervous system in cases of a Cys/GSH deficiency.

Attempts to elucidate the roles of γ-glutamyl peptides in the brain have just begun, and available information concerning them is limited. We therefore discuss this issue using the liver as an example because it is the most extensively investigated organ. During the production of Cys through the transsulfuration pathway, 2-oxobutyrate is produced as a result of the CSE-catalyzed cleavage of cystathionine [228]. The resulting 2-oxobutyrate is converted into 2-aminobutyrate (2AB) by transferring an amino group from Glu via aminotransferases. Because 2AB is also a preferred substrate for γ-GCS next to Cys, γ-Glu-2AB is produced under conditions of a Cys insufficiency. Whereas γ-GCS is suppressed by physiological levels of GSH via a feedback mechanism, the consumption of GSH stimulates γ-GCS activity [65]. When γ-GCS utilizes 2AB as the acceptor substrate, γ-Glu-2AB is produced and is then converted to γ-Glu-2AB-Gly by the GSS reaction, which is denoted as ophthalmic acid (OPT) [229]. The production of excess levels of OPT is observed in the mouse liver under conditions of a Cys deficiency, typically upon an acetaminophen overdose [230]. Starvation of mice causes an insufficient supply of amino acids, including Cys and Met, which also leads to an increase in OPT production in the blood plasma [231]. Intriguingly, several types of γ-glutamyl peptides were reported to be elevated in the blood plasma of patients who are suffering from liver diseases [232]. Since the liver is a central organ for amino acid metabolism, hepatocytes may suffer oxidative stress and result in the consumption of both GSH and Cys. It is conceivable that, under these conditions, activation of the transsulfuration pathway produces 2-oxobutyrate along with Cys. However, Cys is rapidly recruited for GSH synthesis, which consequently stimulates the utilization of 2AB for the production of OPT. NAC is generally used in the treatment of an acute liver injury caused by an acetaminophen overdose. For similar reasons, NAC or its lipophilic derivatives increase the levels of cellular Cys and, consequently, GSH, which then may exert therapeutic effects on neuronal diseases [233].

Ferroptosis is reportedly suppressed by producing not only GSH but also other γ-glutamyl peptides by γ-GCS [108]. The anti-ferroptotic effects of the production of γ-glutamyl peptides other than GSH cannot be explained by the reductive detoxification of lipid peroxides via GPX4. Because the inhibition of γ-glutamyl peptide synthesis elevates cellular Glu levels, the stimulation of the Glu metabolism appears to be the likely mechanism for executing ferroptosis under a Cys insufficiency. This notion is consistent with findings that elevations in electrochemical potential in mitochondria are associated with ferroptosis induced by Cys deprivation [234]. Because actively proliferating cells are more sensitive to ferroptotic stimuli, the activation of Glu-centered carbon metabolism likely produces more radical species, which may stimulate lipid peroxidation reactions and consequent ferroptosis [235]. Meanwhile, the cysteine-sparing effect of taurine has been proposed in hepatocytes [236], although taurine cannot directly compensate for a Cys deficiency. These collective data can also be interpreted to indicate that the formation of γ-Glu-Tau and other γ-glutamyl peptides by either intracellular γ-GCS or extracellular GGT prevents the excitatory cytotoxicity caused by excessive levels of Glu.

### 3.7. Calcium-Sensing Receptor as a Target of γ-Glutamyl Peptides

In addition to GSH, many γ-glutamyl peptides have been detected, and the γ-glutamyl moiety of N-terminal amino acids stabilizes them. The physiological significance of this peptide-specific modification has long been debated. The pharmacological benefit of supplementation of γ-Glu-Cys has recently been proposed under pathological conditions, such as strokes [237], inflammation [238,239,240,241], ALS [242], and ischemia/reperfusion injury [243]. The pharmacological action of γ-Glu-Cys appears to be largely attributable to an increased production of GSH [244,245,246]. Although the direct donation of an electron from γ-Glu-Cys to GPX1 has been also demonstrated [247], the presence of GSS mostly converts γ-Glu-Cys to GSH. Therefore, the use of intrinsic γ-Glu-Cys as the substrate for GPX1 may occur in limited organs such as the kidney, which uniquely contains four times more γ-Glu-Cys than GSH [224].

Extracellular levels pf GSH, GSSG, and mixed disulfides between Cys and GSH (CySSG) can modulate the function of the G-protein-coupled calcium-sensing receptor (CaSR) [248,249]. CaSR is systemically expressed, including in the brain and intestine, and a primary function of CaSR appears to be maintaining extracellular calcium ion levels within a physiological range, 1.1–1.3 mM, by regulating the secretion of the parathyroid hormone [250]. CaSR is abundantly expressed in the circumventricular regions and substantially within the hippocampus, hypothalamus, and striatum. Concerning the roles of CaSR in the neuronal regulation of nutrition, several γ-glutamyl peptides appear to bind CaSR. The extracellular domain of this protein contains a binding pocket for GSH-related compounds [249]. The allosteric activation of CaSR by γ-Glu-Cys was reported to suppress inflammation in colitis model mice [238]. CaSR expressed within the gastrointestinal tract plays roles as a mediator of “kokumi taste” modulation and is responsible for regulating the release of dietary hormones in response to amino acids in the intestine [251,252].

The roles of CaSR in neurons reportedly include neuronal growth, migration, differentiation, and neurotransmission [250]. It has been proposed that CaSR has a critical role in the central neuronal system under pathological conditions, such as ischemia, AD, and in neuroblastomas [253]. The expression of CaSR in inflammatory cells may extend its roles to neuroinflammation [254]. Since soluble Aβ reportedly binds CaSR and leads to neuronal inflammation and cell death [255,256], GSH or other related peptides may be able to exert protective action in the neuronal system through modulating the function of CaSR. Some of the γ-glutamyl peptides that are produced through either the GSH-synthesizing pathway or the GGT-mediated transfer of a γ-glutamyl group may play roles in calcium homeostasis in the neuronal system via modulating CaSR.

## 4. Future Perspectives

GSH exerts pleiotropic actions in the central nervous system as well as other tissues, and its deficiency is associated with the development of various diseases, which include neurological diseases, as discussed and listed in Table 1. In addition to the synthesis and degradation of GSH, γ-GCS and GGT participate in the production of a variety of γ-glutamyl peptides. Whereas their usefulness as a biomarker for disease has been suggested, the molecular mechanism responsible for the function of γ-glutamyl peptides has remained vague. Recent advances in this field have gradually clarified this issue, and the functional regulation of CaSR is one of them. Thus, the usefulness of some γ-glutamyl peptides has actually been proposed, and the elucidation of the molecular mechanisms associated with these peptides may extend their applications.

## Figures and Tables

**Figure 1 ijms-24-08044-f001:**
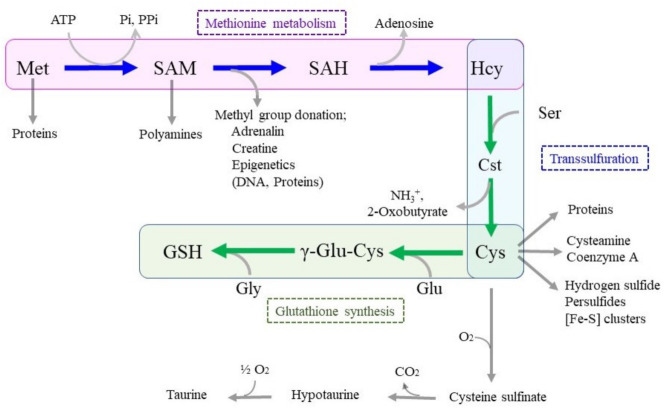
Metabolic linkage of methionine (Met), cysteine (Cys), and glutathione (GSH). Met first reacts with ATP, resulting in the formation of S-adenosylmethionine (SAM). After donation of a methyl group, SAM is converted to S-adenosylhomocysteine (SAH) followed by homocysteine (Hcy) by releasing adenosine. In the transsulfuration pathway, serine (Ser) is ligated to Hcy to form cystathionine (Cst). Cystathionine γ-lyase (CSE) then cleaves cystathionine to Cys and 2-oxobutyrate. Cys is the precursor for not only GSH, but also taurine, cysteamine, and [Fe–S] cluster. Cysteine dioxygenases oxidizes Cys to cysteine sulfonate, which is then converted into hypotaurine by cysteine sulfinate decarboxylase. Hypotaurine is finally converted to taurine by hypotaurine dehydrogenase. Glu, glutamate; Gly, glycine.

**Figure 2 ijms-24-08044-f002:**
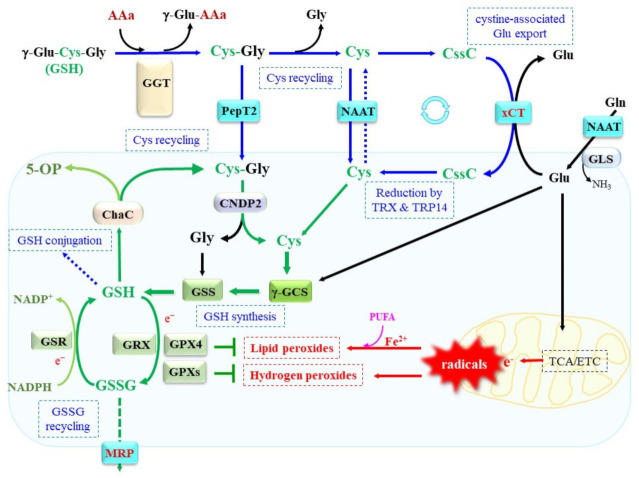
Transporters and enzymes involved in the metabolism of Cys and GSH. The degradation of extracellular GSH is initiated by γ-glutamyl transferase (GGT)-mediated removal of γ-glutamyl group. The resulting Cys–Gly dipeptide is either incorporated directly via the dipeptide transporter PepT2 or degraded to amino acids by the action of extracellular dipeptidases. Cys is incorporated via a neutral amino acid transporter, such as the alanine serine cysteine transporter (ASCT) or neutral amino acid transporter (NAAT), or oxidized to cystine. Cystine is transported into cells via xCT and reduced to Cys by the action of TRX1 and the TRX-related protein 14 kDa (TRP14). γ-GCS ligates Glu to Cys via its γ-glutamyl group, and this results in γ-Glu–Cys. GSS then adds Gly after Cys, which forms GSH. GSH donates an electron to glutathione peroxidase (GPX), which reduces peroxides. GSH is also utilized by glutaredoxin (GRX) to regenerate proteins in reduced states. Glutathione reductase (GSR) reduces oxidized glutathione (GSSG) back to GSH in an NADPH-dependent manner. Degradation of intracellular GSH is initiated by γ-glutamylcyclotransferase (ChaC) that releases 5-oxoproline (5-OP). The remaining Cys–Gly is hydrolyzed to amino acids by cellular dipeptidase, represented by carnosine dipeptidase 2 (CNDP2). Radicals released from mitochondria are major source for peroxides. Notably, GPX4 reduces lipid peroxides in a GSH-dependent manner. CssC, cystine; Gln, glutamine; GLS, glutaminase; PUFA, polyunsaturated fatty acid; γ-GCS, γ-glutamyl–cysteine synthetase; MRP, multidrug resistance regulator; xCT, cysteine transporter.

**Figure 3 ijms-24-08044-f003:**
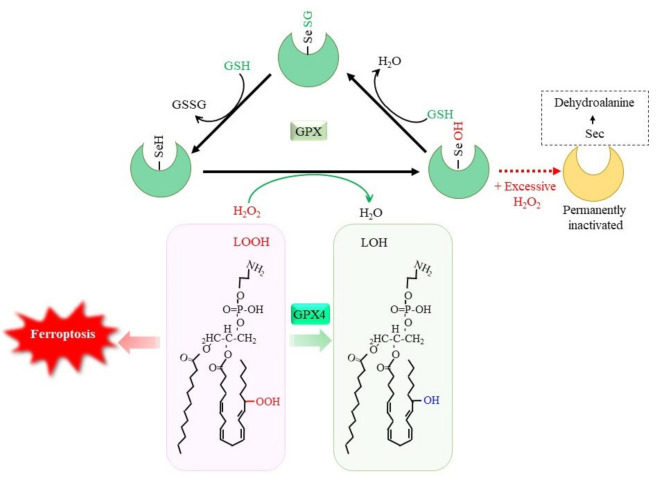
GSH-dependent reduction of peroxides by glutathione peroxidase (GPX). Selenocysteine (Sec) constitutes the catalytic center of GPX1 to GPX4, among GPX family members. The reaction of GPX with peroxides converts SeH to SeOH in the Sec residue, which transiently leads to glutathionylation. Reaction of another GSH regenerates SeH and releases GSSG. Excessive hydrogen peroxide converts Sec to dehydroalanine, leading to permanent inactivation of GPX1. Accumulation of phospholipid hydroperoxides causes ferroptosis, whereas GPX4 specifically reduces them to the alcohol form, leading to cell survival. LOOH, lipid hydroperoxide; LOH, lipid alcohol.

**Figure 4 ijms-24-08044-f004:**
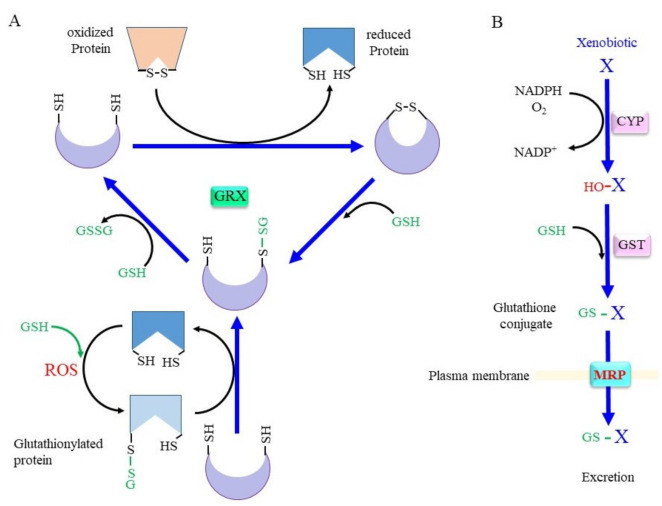
Scheme for the conjugation of glutaredoxin (GRX) and GSH. (**A**) The upper scheme of the reaction shows the reduction of oxidized proteins, and the lower scheme shows GRX accepting the GSH moiety from glutathionylated proteins. When the protein with a disulfide bond reacts with reduced GRX, the protein is reduced instead and GRX is oxidized. Upon reaction with GSH, one cysteine (Cys) residue of the oxidized GRX becomes an SH group and that of the other Cys becomes glutathionylated. Reaction of glutathionylated GRX with another GSH results in fully reduced GRX and GSSG. (**B**) Xenobiotics and hydrophobic metabolite (X) first undergo hydroxylation by oxygenases such as cytochrome P450 oxidases (CYP). Then, glutathione S-transferase (GST) catalyzes GSH conjugation. Resulted glutathione conjugates are exported via multidrug resistance regulator (MRP). ROS, reactive oxygen species.

**Figure 5 ijms-24-08044-f005:**
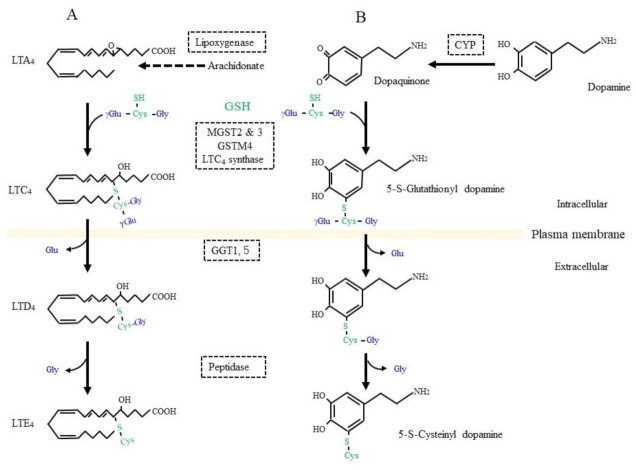
GSH as a building block for some bioreactive compounds. (**A**) GSH is conjugated to leukotriene (LT) A_4_ by the catalytic action of LTC_4_ synthetase or some other glutathione S-transferase (GST) isozyme. γ-Glutamyl transferase (GGT) hydrolytically removes the γ-glutamyl moiety from LTC_4_, which enables access to a peptidase that removes the glycine (Gly) unit from LTC_4_ and results in LTD_4_ formation. (**B**) Dopamine is oxidized to dopaquinone by oxidases such as cytochrome P450 (CYP). After conjugation with GSH, 5-S-cysteinyl dopamine is produced by the action of similar enzymatic process to those for LTD_4_ synthesis.

**Figure 6 ijms-24-08044-f006:**
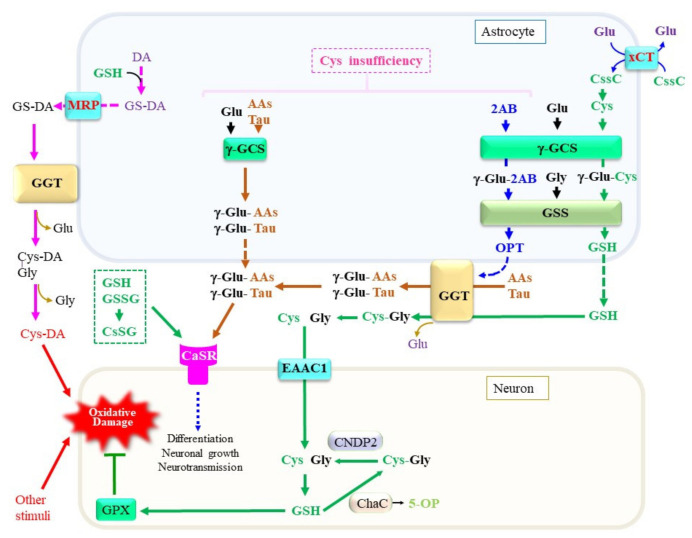
Production, metabolism, and cellular signaling of γ-glutamyl peptides. Due to the diversity of the nervous system, the metabolic pathways shown in this diagram are only examples of a certain central nervous system and may not be applicable to all neuronal cells. When Cys is present at sufficient levels, GSH is produced. However, under a Cys insufficiency, high levels of 2-aminobutryrate (2AB) are synthesized through the transsulfuration pathway, followed by a transaminase reaction, and it becomes a substrate for γ-GCS, which may lead to the production of ophthalmate (OPT). When other amino acids, including taurine, are used instead of Cys, a variety of γ-glutamyl peptides such as γ-Glu-Tau are produced by γ-GCS reactions inside cells. γ-Glutamyl transferase (GGT) on astrocytes either hydrolytically removes the γ-glutamyl moiety of extracellular GSH and γ-glutamyl peptides or transfers their γ-glutamyl moiety to other amino acids (AAs) to generate new γ-glutamyl peptides. The resultant Cys–Gly dipeptide can be hydrolyzed to its constituent amino acids by the action of a dipeptidase. The function of the calcium-sensing receptor (CaSR) may be modulated by the binding of these γ-glutamyl peptides. AAs, any amino acid; DA, dopamine. γ-GCS, γ-glutamyl-cysteine synthetase; GSS, glutathione synthetase; ChaC, γ-glutamylcyclotransferase; MRP, multidrug resistance regulator; CNDP2, carnosine dipeptidase 2; EAAC1, excitatory amino acid carrier 1; GPX, glutathione peroxidase; DA, dopamine; Cys-DA, cysteinyl-dopamine; xCT, cystine transporter.

**Table 1 ijms-24-08044-t001:** A list of cited literatures on GSH in relationship with representative neuronal diseases.

Disease	Description on GSH and Related Subjects	Ref.
Alzheimer’s disease (AD)		
	Overviewing roles of glutathione function	[19]
	Possible involvement of ferroptosis	[199,200,201]
	Protection by glutaredoxin	[169,170]
	Investigation of protective roles of vitamin E	[202]
	Protection by N-Acetylcysteine	[204]
	Roles of the calcium-sensing receptor (CaSR)	[255,256]
Parkinson’s disease (PD)		
	Reviewing roles of glutathione function	[177,185,211]
	Possible involvement of ferroptosis	[206,207,208]
	Toxic roles of dopamine	[209,210,212]
	Roles of DJ-1 in association with glutaredoxin	[124,160]
	MAPK and apoptosis signaling	[164]
	Roles of glutaredoxin	[160,165,167,168]
Amyotrophic lateral sclerosis (ALS)		
	Protection by glutathione	[214]
	Cysteine modifications in the pathogenesis	[215,221]
	Protection by edaravone	[217,218]
	Involvement of ferroptosis	[49]

## Data Availability

Not applicable.

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
