# Peer review of "Critical Roles of the Cysteine–Glutathione Axis in the Production of γ-Glutamyl Peptides in the Nervous System"

_ijms, 2023, doi:10.3390/ijms24098044_

Round 1
Reviewer 1 Report
The review related to the cysteine-glutathione axis in glutathione (GSH) metabolism for neuronal regulation was well-prepared and comprehensive enough to cover the concerning area. The authors utilized a good range of references and separated the headings in a well-arranged manner. There are only a few points that should be mentioned.
I highly recommend to integrate a table summarizing GSH and neuronal disease relationship. There should be disease types, GSH relationship and references.
There can be a paragraph concerning about neuronal development and GSH relationship.
There should be a paragraph related to genotoxicity in neuronal disease and GSH metabolism.
Author Response
Thank you very much for your efforts on reviewing and criticism on our manuscript. We will write our responses after your comments.
1.I highly recommend to integrate a table summarizing GSH and neuronal disease relationship. There should be disease types, GSH relationship and references.
Responses: Thank you for kind advice. Because of importance of GSH, numerous studies have shown association of GSH with neurological diseases. So we described common or representative neurological disorders, Alzheimer’s disease, Parkinson’s disease (PD) and amyotrophic lateral sclerosis (ALS). According to the advice, we have summarized literature in Table 1 that lists GSH in relationship with representative neuronal diseases mentioned in the text (page 18).
2. There can be a paragraph concerning about neuronal development and GSH relationship.
Response: Thank you for another advice on the neuronal development. We consider description on involvement of glutathione in neuronal development is a bit out of focus of this review article. Glutathione is a very important molecule, and its maintenance is necessary for not only neurons but also virtually all cells during development and health. Insufficient synthesis of GSH rather fatal damage to cells, as mentioned in the mice with defected GSH synthesis. Accordingly, we have not included the topic regarding neuronal development and GSH relationship. We have actually discussed the cell death ferroptosis caused by glutathione deficiency, which may occur in neuronal cells during development.
3. There should be a paragraph related to genotoxicity in neuronal disease and GSH metabolism.
Responses: Thank you for another advice again. Regarding genotoxicity, it is often thought to cause tumorigenesis. GSH deficiency may directly associate with detoxification of genotoxic compounds, however, this is mainly performed by the liver or kidney. In case of most other tissues, ROS is the likely genotoxicant. Oxidative DNA damage is caused by elevated ROS, notably hydroxyl radicals, which is the likely event due to a decrease in GPX activity. Because the genotoxic effects of ROS have been discussed in numerous literature, we have added following description on this issue. (Page 10)
“The decline in the GPX activity increases peroxides, which, in the presence of iron, result in production of hydroxyl radicals that cause oxidative DNA damage. Then, cells with DNA damage may undergo tumor development [27]. Oxidative stress-induced tumorigenesis is a major research subject in itself and discussed repeatedly, so we will not discuss it further in this paper.”
Reviewer 2 Report
This review is clearly structured according to the arguments conveyed by the author. It also describes the current status and issues well, and this reviewer do not raise any concerns about the content.
Although not a critical issue, many of the figures seem to have a variety of fonts and sizes, and I found it difficult to read, especially the thin text.
Fonts placed in a colored background tend to be difficult to read, so perhaps a color change or a background color for the fonts could be added.
Author Response
Thank you very much for your efforts on reviewing and criticism on our manuscript. We will write our responses after your comments.
Although not a critical issue, many of the figures seem to have a variety of fonts and sizes, and I found it difficult to read, especially the thin text.
Fonts placed in a colored background tend to be difficult to read, so perhaps a color change or a background color for the fonts could be added.
Response:
Thank you very much for kindly advising us. We dared to change the font and size for according to the importance. Now, we have tried to fix them as much as possible in revised figures.
Reviewer 3 Report
The review “critical roles of the cysteine-glutathione axis in the production of γ- 2 glutamyl peptides in the nervous system” is a very interesting synthesis work dealing with a very important subject.
the review is well-structured with a logical succession of information
The author has produced a very good synthesis that can represent a scientific reference in this field.
I just have a few remarks:
- in the figures, the author has to give the full names of the abbreviations in the legend
- the author should add the molecular regulation in the anabolism and catabolism of GSH and GPX, especially with the signaling pathways.
- the author has to add the regulation of the GSH and GPX following oxidative stress and the relationship with the other enzymes and markers (SOD, Catalase, MDA)
The author has to show the minor revisions in the text, with a different color text, by highlighting the changes.
Author Response
Thank you very much for your efforts on reviewing and criticism on our manuscript. We will write our responses after your comments.
1. in the figures, the author has to give the full names of the abbreviations in the legend
Responses: Thank you for pointing out. We omitted the full names of the abbreviation for terms that have abbreviations in the text. Now we have added them to the legends for readers convenience
2. the author should add the molecular regulation in the anabolism and catabolism of GSH and GPX, especially with the signaling pathways.
Responses: Thank you for critical comments.
Actually, whole session of “2. Regulation of cellular GSH content by balancing synthesis, recycling, and degradation” describes regulation of GSH synthesis and metabolism. In case of GPX, Contents of GSH is the major determinant of the GPX activity. We have mentioned this point as follows (page 9). “Because GPX is constitutive active within cells and requires GSH other than substrate for activity, supply of GSH determines the GPX activity.”
3. the author has to add the regulation of the GSH and GPX following oxidative stress and the relationship with the other enzymes and markers (SOD, Catalase, MDA)
Responses: Thank you for critical comments. As responded above, GPX is constitutive active and mainly regulated by GSH.
We have actually mentioned relationship between GPX and catalaes (page 9) as “While certain other proteins, such as catalase and peroxiredoxin, exhibit peroxidase activities, GPX is a dominant peroxidase family that is made up of 8 members in mammals.”
We have not mentioned “MDA” in this article because MDA is a kind of lipid peroxidation product but cannot react with SOD, GPX, or catalase. So that MDA has no direct association with glutathione metabolism.
4.The author has to show the minor revisions in the text, with a different color text, by highlighting the changes.
Responses: Thank for your adivice. We have color coded the changes in the text and Table 1.
Round 2
Reviewer 1 Report
The authors made a significant effort to address my points. I agree to accept the manuscript in its present form.